# Handwritten Numeral Recognition Integrating Start–End Points Measure with Convolutional Neural Network

**M. A. H. Akhand** [1,*] **, Md. Rahat-Uz-Zaman** [1] **, Shadmaan Hye** [1] **and Md Abdus Samad Kamal** [2,*]

1  Department of Computer Science and Engineering, Khulna University of Engineering & Technology, Khulna 9203, Bangladesh
2  Graduate School of Science and Technology, Gunma University, Kiryu 376-8515, Japan
*  Correspondence: akhand@cse.kuet.ac.bd (M.A.H.A.); maskamal@gunma-u.ac.jp (M.A.S.K.)

**Abstract:** Convolutional neural network (CNN) based methods have succeeded for handwritten numeral recognition (HNR) applications. However, CNN seems to misclassify similarly shaped numerals (i.e., the silhouette of the numerals that look the same). This paper presents an enhanced HNR system to improve the classification accuracy of the similarly shaped handwritten numerals incorporating the terminals points with CNN's recognition, which can be utilized in various emerging applications related to language translation. In handwritten numerals, the terminal points (i.e., the start and end positions) are considered additional properties to discriminate between similarly shaped numerals. Start–End Writing Measure (SEWM) and its integration with CNN is the main contribution of this research. Traditionally, the classification outcome of a CNN-based system is considered according to the highest probability exposed for a particular numeral category. In the proposed system, along with such classification, its probability value (i.e., CNN's confidence level) is also used as a regulating element. Parallel to CNN's classification operation, SEWM measures the start-end points of the numeral image, suggesting the numeral category for which measured start-end points are found close to reference start-end points of the numeral class. Finally, the output label or system's classification of the given numeral image is provided by comparing the confidence level with a predefined threshold value. SEWM-CNN is a suitable HNR method for Bengali and Devanagari numerals compared with other existing methods.

**Keywords:** classification; convolutional neural network; handwritten numeral recognition; start-end writing measure

## 1. Introduction

Numerals are an integral part of any language and play an essential role in everyday life. Currently, numerals are used in printed and handwritten forms in everyday activities and business. Handwritten numerals are used in postal codes, bank cheques, and many other businesses [1]. Automatic recognition of handwritten numerals can release tedious jobs at banks, post offices, and registration departments. Therefore, handwritten numeral recognition (HNR) has become an important research topic. HNR poses more significant challenges than character or word recognition. In the case of character or word recognition, recognition errors made by a system can be verified using the rules of grammar. However, such an error detection rule cannot be applied to numeral recognition, since any numeral combination is technically valid. Thus, recognizing handwritten numerals is a sensitive task, and the system must be accurate for individual digits.

Traditionally, HNR is considered a pattern recognition task that includes pre-processing of handwritten numeral images, extraction of features, and classification of the images into different numeral categories. Principal component analysis (PCA) [2], genetic algorithm (GA) [3], Bayes theorem, maximum a posteriori and k-means clustering [4], local binary patterns [5], histogram of oriented gradients [6,7], convex hull [8], chain code and

Fourier descriptor [9], wavelets [10–12], GIST descriptor [13], etc., are used for feature extraction. Then, different classification algorithms like support vector machine (SVM) [2,3,7], K-nearest neighbor (KNN) [14], naïve Bayes, random tree [15], random forest [16], etc., are applied to classify the images into different numeral categories.

Recently, convolutional neural network (CNN) has been used most frequently for image analysis and classification, including HNR [17–22]. CNN does not require any separate feature extraction step, as it can extract the inherent features from the image data through its deep layered structure. A number of CNN-based studies are shown to outperform the other methods for HNR [22–29]. However, CNN-based methods show unsatisfactory performance for similarly shaped patterns [22]. In the case of similarly shaped numerals (i.e., the silhouette of the numerals that look the same), CNN cannot distinguish between these handwritten numerals.

Challenges of HNR are language dependency based on shape, similarity, and other complexities in the numeral sets. When a language contains similarly shaped numerals, recognition becomes difficult even for humans because the similarities turn out to be very close due to variations of writing patterns of people. On the other hand, the same numeral may look very different in size, shape, and orientation due to different writing patterns. Therefore, numerals with similarly shaped patterns have lower recognition accuracy than other languages. Among the major languages, Bengali and Devanagari suffer from low recognition accuracy due to the similarly shaped numerals. Therefore, HRN system development focusing on similarly shaped patterns is an open research challenge.

The main objective of this work is to build a novel HNR system integrating features of human writing style with the existing pattern recognition technique of CNN. Such an HNR can be used in various emerging applications, including language translation, voice conversation from the typical handwritten text of any language. A hypothesis behind the idea is that the writing direction and style of a particular numeral in a language are common because people learn to write numerals by practicing on the particular pattern. For example, the writing direction of the Bengali numeral ১ is from top to bottom, but it is bottom to up for the numeral ৯ in general, even though they look similar. Therefore, when CNN confuses classifying a numeral image between ১ and ৯, the inclusion of the writing direction would be a good choice for classification. The proposed method consists of three important stages: classification of a given numeral image using standard CNN, start-end points measure of the numeral, and the final classification decision integrating the start-end points measure with CNN's decision. Therefore, the significant contribution of the present study is the start-end points measure from a handwritten numeral image and integration of such measures for better HNR recognition. For a fair evaluation of the proposed method, we consider both Bengali and Devanagari numerals, as the HNR accuracy of these languages is relatively low due to inherited challenges. For both types of numerals, the recognition efficiency and various performance measures are compared with the proposed HNR. The major contributions of this research are summarized in the following points:

- This study proposes an improved HNR method focusing on similarly shaped numerals;
- Start-end points measure of a handwritten numeral image is integrated into the decision of CNN for HNR recognition.
- Enhanced recognition accuracy demonstrates the superiority of the proposed system over existing methods.

The rest of the paper is organized as follows: Section 2 describes the proposed system architectures along with the dataset description. Section 3 investigates the efficiency of the proposed method through experimental results and analysis. The section also compares the performance of the proposed method with other related works. Finally, a brief conclusion of the work is given in Section 4.

## 2. HNR Integrating Start-End Writing Measure with CNN (SEWM-CNN)

Although CNN-based methods outperformed other existing methods of HNR, they often misclassify similarly shaped numerals [22] as discussed in the above section. According to a recent study [22], handwritten images of ১ (i.e., 1) and ৯ (i.e., 9) of Bengali numerals are interchangeably misclassified in large cases regarding other numerals. The confidence level of CNN is found to be relatively low as well as similar for such numerals. In order to enhance the classification accuracy of similarly shaped numerals, the start-end points measure is introduced as a new feature for classification in addition to CNN. Figure 1 demonstrates the proposed HNR system integrating the Start–End Writing Measure with CNN (SEWM-CNN). Handwritten numeral images may look very different in size, shape, and orientation due to the different writing patterns of individuals. Therefore, the proposed SEWM-CNN is considered a pre-processing step like any image-based recognition system to make the images similar sized to put in the system.

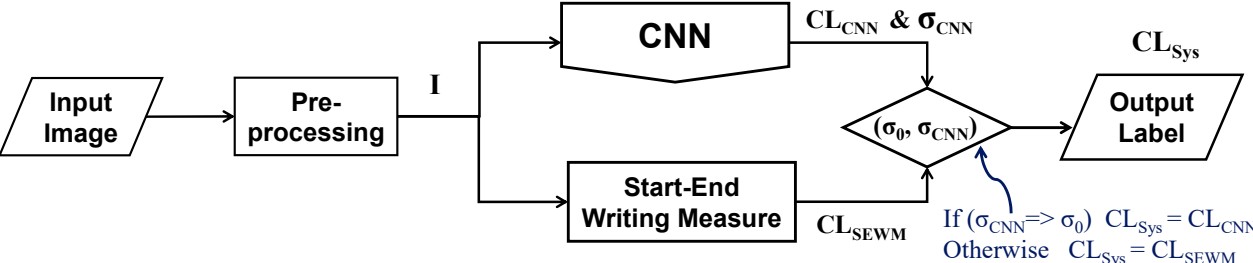

**Figure 1.** Structure of the proposed HNR method integrating Start–End Writing Measure with CNN.

There are three major functional steps to classify the pre-processed numeral images (I) in the proposed SEWM-CNN: classification using a CNN; identification of start and end writing points; and finally, system output integrating start-end points measure with the CNN decision. The outcome of CNN is the probability of the input numeral image into different numeral categories. Traditionally, CNN'S classification (i.e., $CL_{CNN} \in \{0, 1, \ldots, 9\}$) for the highest probability is the outcome of a CNN-based system. In the proposed system, along with classification in $CL_{CNN}$, its probability value (say CNN's confidence level $\sigma_{CNN}$) is also considered as a regulating element. Parallel to CNN's classification operation, SEWM measures the start-end points of the numeral image, suggesting the $CL_{SEWM}$ numeral category for which measured start-end points are found close to reference start-end points of the numeral class. Finally, the output label or system's classification ($CL_{Sys}$) of the given numeral image is provided by comparing $\sigma_{CNN}$ with a predefined threshold value ($\sigma_0$): $CL_{Sys} = CL_{CNN}$ if $\sigma_{CNN} => \sigma_0$; otherwise, $CL_{Sys} = CL_{SEWM}$. The following subsections describe the pre-processing of data and the three functional steps of the system.

### 2.1. Dataset Selection and Pre-Processing

Bengali and Devanagari numerals are chosen to verify the proposed method because both the scripts contain similarly shaped numerals, and hence accuracy could be made higher despite challenges in classifying such numerals. Another essential factor for selecting the scripts is their extensive usability; both are major scripts in the south Asia region, and a vast population worldwide uses them. Among several collections of handwritten numerals of both the scripts, and datasets of the Computer Vision and Pattern Recognition Unit, Indian Statistical Institute (ISI) [30] is the most prominent one and has been used in several recent studies as a benchmark. ISI datasets contain a relatively large number of training and test samples for both Bengali and Devanagari scripts. The samples are from postal codes written by different people of different sex, age, and educational level. The Bengali dataset holds 19,392 training image samples and 4000 testing image samples. The training and test sets of the Devanagari dataset contain 18,793 and 3763 image samples, respectively. In this study, 18,000 (=1800 × 10) images are used for training purposes for each Bengali and Devanagari numeral. On the other hand, all the available test samples

are employed for both Bengali and Devanagari. Table 1 shows several images from every numeral of both Bengali and Devanagari, which reveals the level of ambiguity and challenges in recognition. It is easily visible from the presented images that several images from different numeral categories look very similar in shape, including samples of ১ with ৯ in Bengali and samples of ४ with ५ in Devanagari.

**Table 1.** Handwritten numeral samples from ISI Bengali and Devanagari datasets.

| English Numeral | Bengali Numeral | Sample Bengali Handwritten Numeral Images | | | | | Devanagari Numeral | Sample Devanagari Handwritten Numeral Images | | | | |
|---|---|---|---|---|---|---|---|---|---|---|---|---|
| 0 | ০ | | | | | | ० | | | | | |
| 1 | ১ | | | | | | १ | | | | | |
| 2 | ২ | | | | | | २ | | | | | |
| 3 | ৩ | | | | | | ३ | | | | | |
| 4 | ৪ | | | | | | ४ | | | | | |
| 5 | ৫ | | | | | | ५ | | | | | |
| 6 | ৬ | | | | | | ६ | | | | | |
| 7 | ৭ | | | | | | ७ | | | | | |
| 8 | ৮ | | | | | | ८ | | | | | |
| 9 | ৯ | | | | | | ९ | | | | | |

Pre-processing is an essential task of feeding numeral images into the recognition system, and a simple pre-processing is employed in this study. The image samples in both datasets are of various sizes, resolutions, and shapes. Automatic thresholding is applied to the images to generate binary images. The foreground and background are interchanged to reduce the numeral values because the original numerals are written in black on white background leading to more values of 1. Once the foreground–background interchange is performed, there are fewer ones and more zeros, which reduces the computational overhead. Then, all the images are resized to the size of $28 \times 28$.

## 2.2. Related Studies with ISI Datasets

There are some well-known HNR works available using ISI datasets. The pioneering work with the datasets is reported in [31], where wavelet filter-based selected features are used in a cascade of four multi-layer perceptions (MLPs) for classification. For Devanagari HNR, the works [32,33] and [34] utilized samples from the ISI Devanagari dataset. Recently, Guha et al. [35] developed an HNR system using a Memory-Based Histogram with GA for feature selection and KNN for classification; and the method is tested on selected samples from both Bengali and Devanagari datasets. On the other hand, several CNN-based methods have also used the datasets as a benchmark. Akhand et al. [23] re-

ported pioneering work with CNN for Bengali and Bengali–English mixed HNR using the ISI Bengali dataset. Shopon et al. [36] also investigated auto-encoder (AE) with CNN for Bengali HNR. Recently, Akhand et al. [22] developed two different CNN-based models with rotation-based generated patterns from available numeral images and tested them on both Bengali and Devanagari datasets.

### 2.3. Classification with CNN

The purpose of a classifier is to assign each of its admissible inputs to one of the finite numbers of classes by computing a set of decision functions. For two-dimensional data such as images, CNN [37] performs well with its convolution and subsampling mechanisms which can capture rotations, shift invariance, and scale. A standard CNN model of HNR [22,38] with the following architecture is used in the proposed SEWM-CNN.

$$I_{28\times28} \rightarrow \{6K1_{5\times5}C1_{24\times24} - S_{2\times2}6S1_{12\times12}\} \rightarrow$$

$$\{12K2_{5\times5}C2_{8\times8} - S_{2\times2}12S2_{4\times4}\} \rightarrow \{Wo_{192\times10}\} \rightarrow O_{10}$$

The CNN has two convolution-subsampling layers and a fully connected layer, and Figure 2 depicts the structure. The input (*I*) is a pre-processed image with a size of 28 × 28. The first convolution operation with six 5 × 5 sized kernels (*K1*) on *I* produces 24 × 24 sized six different convolved feature maps (CFMs) of C1. Then, subsampling is used to half the width and height of each CFM with a pooling area of 2 × 2, and the outcomes are six 12 × 12 sized sub-sampled feature maps (SFMs) in S1. The second convolution operates 12 kernels (*K2*) of size 5 × 5 on S1, and outcomes are 8 × 8 sized 12 CFMs in C2. Again, the second subsampling operation with a 2 × 2 pooling area produces 4 × 4 sized 12 SFMs in S2. The 192 (=12 × 4 × 4) values of these 12 SFMs are placed linearly as a hidden layer (*H*) with 192 individual nodes. Finally, nodes of *H* are fully connected to the output layer (*O*) through *Wo.* The output layer contains 10 nodes, and a particular node represents a particular numeral class. The desired value in a particular node is 1 (and the other 9 output nodes value as 0) for the input of a particular numeral category. Training of CNN is performed to obtain appropriate values of kernels (*K1* and *K2*) and weight (*Wo*) values so that it correctly recognizes a test image generating 1 (or close to 1) in the correct output node. Output values generated in 10 output nodes are normalized (to sum as 1) to obtain the classification probability in individual numeral classes. The highest probability value (as confidence level $\sigma_{CNN}$) and its class category ($CL_{CNN}$) are the outcomes of CNN, which are proceeded for the system's outcome. The detailed training operation of CNN, along with a description of its structure, is available in [22,38].

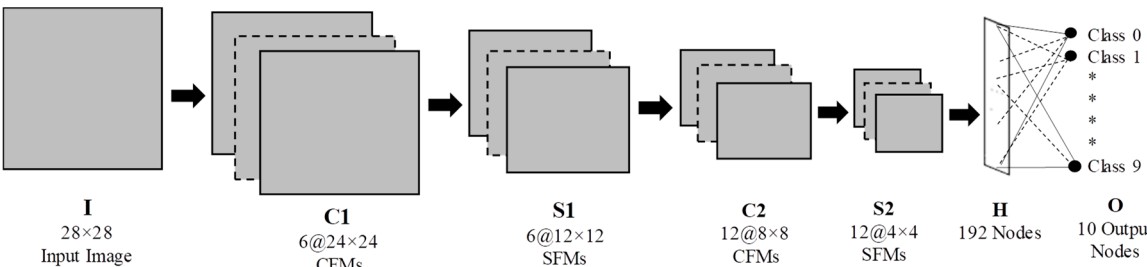

**Figure 2.** Structure of CNN used in the proposed SEWM-CNN method for HNR.

### 2.4. Start–End Writing Measure from Numeral Image

Integrating the Start–End Writing Measure (SEWM) with CNN to improve the accuracy of the HNR system is the main contribution of this study. Handwritten numerals are the outcomes of sequences of strokes on paper using a pen or similar writing device. Start and end points can make distinguishable features of numerals while they look similar in shape. For example, images of Bengali numerals ১ and ৯ look similar to each other, but the start and end positions are just opposite to each other; that is, the start position of the

numeral ؟ is at the top while the start position of the numeral ؟ is at the bottom. Since the close similarity between handwritten numerals is a well-established source of low recognition accuracy, including the start-end positions of writing to the feature set would be an effective measure to resolve the ambiguity posed by a primary classifier such as CNN. It is worth noting that a numeral comprises a sequence of single strokes. However, finding the start and end positions from an image is a challenging task. The following sections describe the extraction method of start-end writing positions in individual numeral images and determining numeral-wise start-end reference points.

Extraction of Start-End Positions in a Numeral Image

The steps to find start-end points of writing in a numeral image are extraction of the skeleton through thinning and then identification of the terminal points. Thinning is actually the transformation of a numeral image so that the width of the stroke sequence is only one pixel. The Zhang-Suen [39] thinning algorithm is applied for this purpose. The algorithm works on binary images and iteratively turns a pixel black if it is a white boundary pixel until reaching a terminating condition. The Zhang-Suen algorithm is a popular thinning algorithm, and a detailed description of the algorithm is available in [39,40].

A traveling salesman problem (TSP) algorithm is applied to the skeleton of the image for stroke sequence extraction. In traditional TSP, a graph of cities (i.e., a map) is given, and the salesman is required to visit every city only once, starting from an initial city and returning to the starting city [41]. The main task of the TSP algorithm in this study is to establish the connectivity among the pixels owing to finding writing start-end points. In traditional TSP cases, any node may consider a starting point as it is back to the starting point. On the other hand, the significant action of sequence generation in the HNR case is seeking the sequences to start from the top left endpoint to ensure consistent transformation. However, the endpoints might be in different positions for different numerals and samples. In TSP operation for stroke sequence extraction, the white pixels are considered as nodes of the TSP, and the distance is measured with the Euclidean distance function. As neighboring pixels are the closest nodes, they will be visited sequentially by the writing device (e.g., pen). The outcome of the TSP algorithm is a sequence from the start point to the endpoint or vice versa. Actual start and end positions are not found in the TSP operation. Therefore, additional action is introduced to identify the actual start and end points of a numeral image.

An innovative idea based on human writing behavior is utilized to determine the actual start and end points of the sequence generated from the skeleton of a numeral image. Specifically, pen stroke characteristics to the sequence are considered a realistic process of finding the start and end of writing. While writing, people generally put higher pressure at the starting point and release the pressure at the ending point. Therefore, the width of pen strokes at the starting point is larger than at the endpoint and is verified in this study for considered benchmark numeral images. Therefore, the widths of two endpoints (given by TSP sequencing) are measured from the original image, and the wider end is considered as a writing start point, and another end is considered as the end of writing. Figure 3 demonstrates the whole process of writing start (* marked) and end (# marked) points identification on a sample image with individual stepwise outcomes.

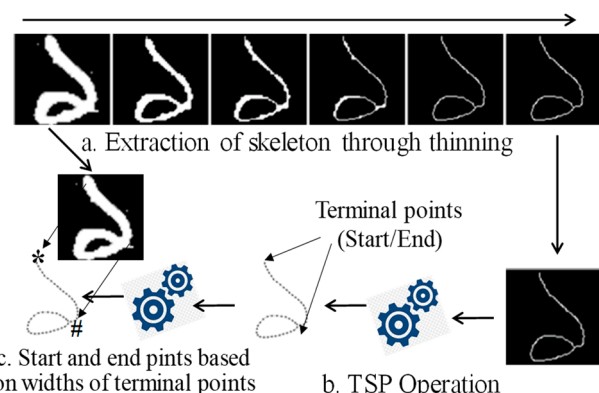

**Figure 3.** Demonstration of writing start (* marked) and end (# marked) points identification on a sample image.

Determining Numeral-Wise Reference Start-End Points

Individual numeral-wise reference start-end points are essential to distinguish the numerals on the basis of start and endpoints. The start and end points of a numeral vary among the different handwritten numeral images due to the writing variation of individuals. The average position of individual numerals' start (and end) points is an easy way to obtain the reference start (and end) points of a particular numeral. However, the technique is identified as less meaningful because misidentification of the start as the end (and vice versa) of a few handwriting cases leads to moving the reference point toward an inappropriate position.

A clustering-based technique is chosen for finding the numeral-wise reference of start and endpoints. Different individuals follow different writing styles; thus, the number of clusters for the start (and end) for a particular numeral may be more than one, vary in number, and even be unknown. Therefore, the Density-based Spatial Clustering of Applications with Noise (DBSCAN) [42,43] algorithm is employed in this study where the spatial position of each start or end point and proximity of the points is the key to the clustering. Reference start and end points are marked on the printed numeral images, which are two points in a 2D pixel space.

Reference start and end points measurement is demonstrated for ISI Bengali and Devanagari datasets in Figure 4. Three left columns of the table are for Bengali numerals: start-end points marking are shown on two sample handwritten images in columns 1–2, and finally, DBSCAN cluster-based reference points for start-end points on the printed image are shown in column 3. For marking, star (*) in red is used for the start point, and hash (#) in blue is used for the endpoint. Notably, start and end reference points are almost the same for Bengali numerals ০, ৪, and ৫, as those are closed-loop writing numerals, and hence start and end points marked are overleaped. For other numerals such as ১, ২, ৩, ৬, ৭, ৮, and ৯, the start and end reference points are different, and more interestingly, reference points of start and end points for ১ and ৯ are on opposite locations, though they look very similar in shape. Similar cases are presented for Devanagari in columns 4–6.

**Figure 4.** Numeral-wise start and end points marking on sample images; and start and end reference points marking on printed numerals.

Individual numeral-wise reference start-end points are used to identify for which numeral a given numeral image is close. At first, start and end points are calculated from the given image following the previous steps. Then, Euclidian distances are measured with reference start and end points of each numeral. The numeral with the smallest distance is assigned to the test numeral. Suppose $(S_i, E_i)$ is the calculated start and end positions for the given image; and the reference start and end position pairs of 10 numerals are $(S_0, E_0)$, $(S_1, E_1)$, - - - -, and $(S_9, E_9)$. The distance with 0 numeral is:

$$D_0 = (DS_{i,0} + DE_{i,0})/2, \tag{1}$$

where $DS_{i,0}$ (and $DE_{i,0}$) are the Euclidian distance of the start (and end) points of the given image and reference points of numeral 0. Similarly, distances with other numeral references are $D_1, D_2, \ldots$ , and $D_9$. The outcome of the SEWM (i.e., $CL_{SEWM}$) is the numeral for which distance is minimum, i.e., $min\{D_1, D_2, \ldots D_9\}$.

## 2.5. System Outcome Considering CNN's Confidence and Start–End Writing Measure

The final decision on recognition of the proposed SEWM-CNN depends on the CNN's confidence level ($\sigma_{CNN}$) in the classification of the input image. Suppose with $\sigma_{CNN} => 0.7$, classifying with such a high confidence value assures very low confidence levels of other categories, and the sum of those is 0.3 or less. On the other hand, classification with $\sigma_{CNN} <= 0.5$ indicates that cthe onfidence level in any other category might be competitive. Competitive (as well as low confidence) in two different numeral categories for a numeral image

indicates that CNN becomes confused when making a decision. Such a scenario is typical for similarly shaped numerals. In such a case, the decision from SEWM is considered as the final recognition category of the proposed SEWM-CNN system.

In short, obtaining the SEWM-CNN output is a selection process between the decisions of CNN and SEWM, as shown in Figure 1. The selection is performed based on a defined threshold value ($\sigma_0$). The final system outcome will be CNN's recognition category (i.e., $CL_{Sys} = CL_{CNN}$) if it classifies the image with a confidence level equal to or above $\sigma_0$ (i.e., $\sigma_{CNN} => \sigma_0$). On the contrary, the outcome of SEWM is exposed as a system outcome (i.e., $CL_{Sys} = CL_{SEWM}$) for $\sigma_{CNN} < \sigma_0$. It is notable that such selection-based integration does not incur computational costs in system operation concerning CNN or SEWM.

Suppose the probability values of a sample image in 10 numeral classes by CNN are [0, 0.54, 0, 0, 0, 0, 0, 0, 0, 0.46] and distances between start-end points of the numeral image with reference start-end points of individual numerals from SEWM are [13.3, 19.4, 22.0, 11.4, 13.8, 17.5, 14.0, 21.7, 7.7]. Therefore, $CL_{CNN} = 1$ with $\sigma_{CNN} = 0.54$ and $CL_{SEWM} = 9$ for the lowest distance of 7. If $\sigma_0 = 0.6$ (i.e., $\sigma_{CNN} < \sigma_0$), the system relies on SEWM and $CL_{Sys} = CL_{SEWM} = 9$. It seems the value of $\sigma_0$ between 0.5 and 0.8 is suitable in SEWM-CNN.

### 2.6. Significance of the Proposed System

The primary significance of the present study is developing a novel HNR system based on the hypothesis of human writing style; the proposed method is already demonstrated in Figure 1 and described as well. There is a significant difference between the proposed SEWM-CNN and the traditional techniques for HNR. Diverse writing styles produce very closely similarly shaped numerals; thus, the image-based method CNN becomes confused in recognizing such numeral images in an appropriate category. Another numeral writing hypothesis is integrated with this study to strengthen the recognition ability. The hypothesis of start-end writing positions of individual numerals is considered an additional technique to distinguish such similarly shaped numerals. Start-end points measured from a numeral image is a challenging task, and different innovative steps are taken into account for the task. While existing CNN-based methods considered changes in CNN architecture, use several CNNs (i.e., ensemble), or apply data augmentation to improve performance, the proposed method is entirely different and employs an innovative approach to achieve better performance with CNN.

Integration of SEWM with CNN is also significant in terms of computational cost, although the computational cost is not so important due to the easy availability of high computing machines nowadays. In the proposed system, CNN training and numeral-wise reference start-end points identification are the main computational tasks, and these two tasks are independent of one another. Reference start-end points identification took less computational operation than CNN training on certain iterations. More importantly, the integration of SEWM does not incur computational costs in system operation. A system with SEWM is obviously computationally effective concerning other existing approaches to enhance CNN's performance. The well-performed CNN-based method [22] followed a kind of data augmentation that seems three times computationally heavy with respect to a single CNN.

## 3. Experimental Studies

This section verifies the proposed SEWM-CNN method on the ISI datasets of Bengali and Devanagari scripts. The descriptions of the datasets are already given in the previous section. The performance of the proposed method is investigated for different important issues, such as varying threshold values ($\sigma_0$) of the system in decision-making between CNN and SEWM. Finally, the proficiency of the proposed SEWM-CNN is validated by comparing it with other prominent methods.

The proposed system was implemented in python using Keras and Tensorflow. The experiments were carried out onto Windows 10 OS and Python 3.6 Anaconda environment

on a Desktop PC with the following configuration: Intel(R) Core i7-7770 CPU @ 4.20 GHz, 16 GB RAM, and Nvidia GTX 1050ti 8GB GPU.

### 3.1. Experimental Results and Analysis

The training and test samples are available separately in ISI datasets. Training samples are used to train the CNN and measure the start-end reference points of individual numerals; samples of the test set are used to measure the final efficiency (i.e., generalization ability) of the system. Training is conducted with different batch sizes (BS), as the number of samples in a training batch affects the system's performance. On the other hand, the σ0 value is an important parameter of SEWM-CNN. Therefore, the performance is measured against different values. Experimental outcomes of standard CNN (i.e., CNN alone) are also considered to realize the effect of SEWM integration with CNN in the proposed SEWM-CNN.

Table 2 shows the test set recognition accuracy for Bengali and Devanagari with CNN alone and the proposed SEWM-CNN varying BS from 8 to 128. Due to batch size variation, recognition accuracy is shown to vary for both CNN and SEWM-CNN for Bengali and Devanagari. Moreover, smaller batch size values showed better recognition accuracy for both scripts. For a small batch size, CNN is updated considering a relatively smaller number of training samples at a time. How the performance improves with CNN training is shown in Figure 5 for Bengali for a batch size of 16. It is observed from the figure that at the beginning (iteration up to 20), the recognition accuracy is low, which improves with iteration for both CNN and SEWM-CNN with any $\sigma_0$ value. It is notable that SEWM-CNN consistently outperformed CNN, and SEWM-CNN with $\sigma_0 = 0.6$ is shown to be the best. Nevertheless, accuracy declines after 80 iterations for any case, indicating overfitting. The best recognition accuracy of a method is considered and compared in Table 2.

**Table 2.** Test set recognition accuracy of CNN and proposed SEWM-CNN with different threshold values for different batch sizes.

| (a) Bengali | | | | | |
|---|---|---|---|---|---|
| Batch Size | Recognition accuracy (%) of CNN | Recognition accuracy (%) of proposed SEWM-CNN with different $\sigma_0$ values | | | |
| | | 0.5 | 0.6 | 0.7 | 0.8 |
| 8 | 98.98 | 99.13 | 99.20 | 99.10 | 99.03 |
| 16 | 98.98 | 99.03 | 99.20 | 99.03 | 99.0 |
| 32 | 98.60 | 98.70 | 98.85 | 98.83 | 98.65 |
| 64 | 98.58 | 98.68 | 98.80 | 98.73 | 98.60 |
| 128 | 98.43 | 98.58 | 98.65 | 98.60 | 98.48 |
| (b) Devanagari | | | | | |
| Batch Size | Recognition accuracy (%) of CNN | Recognition accuracy (%) of proposed SEWM-CNN with different $\sigma_0$ values | | | |
| | | 0.5 | 0.6 | 0.7 | 0.8 |
| 8 | 99.08 | 99.13 | 99.23 | 99.10 | 99.08 |
| 16 | 99.10 | 99.15 | 99.23 | 99.15 | 99.13 |
| 32 | 98.78 | 98.88 | 98.90 | 98.85 | 98.83 |
| 64 | 98.78 | 98.85 | 98.83 | 98.78 | 98.75 |
| 128 | 98.50 | 98.58 | 98.70 | 98.55 | 98.53 |

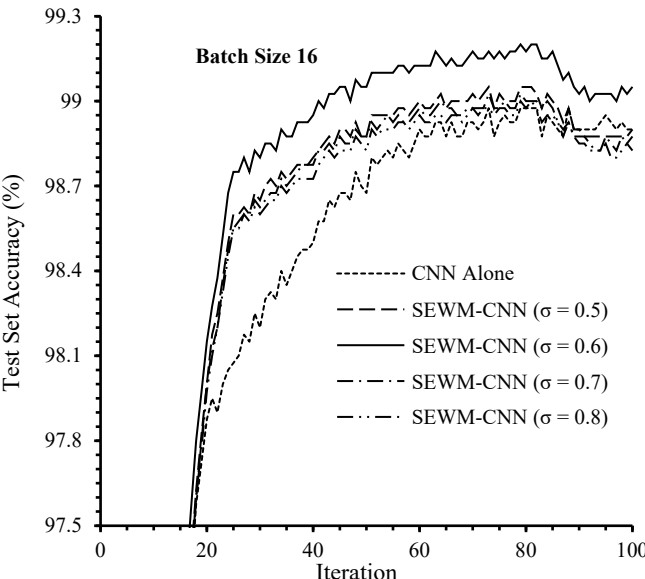

**Figure 5.** Performance of CNN and proposed SEWM-CNN varying iteration on ISI Bengali dataset.

A remarkable observation from Table 2 is that SEWM-CNN outperformed CNN for any BS value, which indicates the proficiency of the proposed approach. As an example, the best recognition accuracy of CNN for Bengali (in Table 2a) is 98.98% for BS values of 8 and 16. On the other hand, for the same BS values, SEWM-CNN achieved a recognition accuracy of 99.20% for $\sigma_0$ = 0.6. The accuracy of SEWM-CNN for any other $\sigma_0$ value (i.e., 0.5, 0.7, or 0.8) is also better than CNN. A similar observation is also available for Devanagari in Table 2b; CNN is shown to have the best recognition accuracy at 99.10% for BS 16; SEWM-CNN with any $\sigma_0$ value achieved better accuracy than CNN. Similar to Bengali, SEWM-CNN achieved the best accuracy with $\sigma_0$ = 0.6, and the value is 99.23%. It is notable that SEWM-CNN considers decisions from the SEWM technique in relatively large numbers for higher $\sigma_0$ values (e.g., 0.8), and similar start-end reference points among several numerals might incur wrong decisions in several cases. On the other hand, for lower $\sigma_0$ values (e.g., 0.5), SEWM-CNN mostly considers CNN's outcome as a system decision ignoring SEWM and hindering the use of the start-end reference measure. According to the results presented in Table 2, $\sigma_0$ = 0.6 is found to be the most suitable value for both Bengali and Devanagari numerals, although all other values are shown to improve the recognition accuracy of the system, rectifying CNN's decision in a range.

Individual numeral-wise recognition analysis for better understanding is presented in Table 3 for both Bengali and Devanagari. For a script (Bengali/Devanagari), the outcomes of CNN and SEWM-CNN with $\sigma_0$ value 0.6 are presented for the same BS value 16. According to Table 3a, there is a total of 41 test samples (out of 4000) Bengali dataset misclassified by CNN, and the number reduced to 32 for SEWM-CNN; hence, recognition accuracy improved from 98.98% to 99.20% as presented in Table 2a. CNN misclassified the Bengali numeral ১ as ৯ in 12 alone out of a total of 18 misclassified cases, as seen in Table 3a. On the contrary, SEWM-CNN misclassified ১ as ৯ in seven cases, and the total misclassified number was reduced to 16. The promising result has been found by SEWM-CNN in the case of ৯; all eight misclassified cases as ১ by CNN are rectified by SEWM-CNN, and the total misclassified number is now reduced from 10 to 2. It is already mentioned that numerals ১ and ৯ are similar in shape, even in printed form, and it is sometimes difficult to distinguish between them. Table 3b also shows a similar observation for Devanagari due to the SEWM total number of misclassified test samples reduced from 36 (out of 3763) to 29; hence, recognition accuracy improved from 99.10% to 99.23%, as presented in Table 2b. In individual numeral cases, interchangeable misclassifications between ४ and ५ and between ६ and ९ are reduced. Finally, the reduction of interchangeable misclassifications of

the numerals clearly indicates the effectiveness of SEWM in improving performance in the proposed SEWM-CNN method.

**Table 3.** Individual numeral-wise performance of CNN and proposed SEWM-CNN on test samples.

| (a) Bengali | | | | | |
|---|---|---|---|---|---|
| Bengali Numeral | Total Samples | CNN | | SEWM-CNN | |
| | | Truly Classified | Misclassification to Other Numerals and Count | Truly Classified | Misclassification to Other Numerals and Count |
| 0 | 400 | 399 | ৫(1) | 399 | ৯(1) |
| ১ | 400 | 382 | ২(1)-৪(1)-৫(2)-৬(1)-৭(1)-৯(12) | 384 | ২(1)-৪(3)-৫(2)-৭(2)-৮(1)-৯(7) |
| ২ | 400 | 398 | ৫(1)−৯(1) | 399 | ৮(1) |
| ৩ | 400 | 400 | - | 399 | 0(1) |
| 8 | 400 | 400 | - | 399 | ৫(1) |
| ৫ | 400 | 394 | ২(2)- ৪(2)-৬(1)-৭(1) | 395 | ১(1)-২(1)-৭(1)-৮(1)-৯(1) |
| ৬ | 400 | 397 | ৩(1)-৫(2) | 398 | 0(1)-১(1) |
| ৭ | 400 | 399 | ৬(1) | 399 | ৪(1) |
| ৮ | 400 | 400 | - | 398 | ২(1)-৩(1) |
| ৯ | 400 | 390 | ১(8)-৬(1)-৭(1) | 398 | ৬(1)-৭(1) |
| Total | 4000 | 3959 | 41 | 3968 | 32 |
| (b) Devanagari | | | | | |
| Devanagari Numeral | Total Samples | CNN | | SEWM-CNN | |
| | | Truly Classified | Misclassification to Other Numerals and Count | Truly Classified | Misclassification to Other Numerals and Count |
| ० | 369 | 364 | ४(1)-७(3)-८(1) | 364 | ४(1)-७(3)-८(1) |
| १ | 378 | 376 | ०(1)-३(1) | 376 | ०(1)-३(1) |
| २ | 378 | 376 | १(1)-५(1) | 376 | १(1)-५(1) |
| ३ | 377 | 375 | ६(1)-९(1) | 374 | ५(1)-६(1)-९(1) |
| ४ | 376 | 372 | ०(1)-५(3) | 374 | ०(1)-५(1) |
| ५ | 378 | 370 | ३(1)-४(7) | 374 | ३(2)-४(2) |
| ६ | 374 | 367 | ३(1)-८(1)-९(5) | 370 | ३(1)-८(1)-९(2) |
| ७ | 378 | 377 | ०(1) | 377 | ०(1) |
| ८ | 377 | 376 | ५(1) | 374 | ५(2)-९(1) |
| ९ | 378 | 374 | ६(4) | 375 | ६(2)-८(1) |
| Total | 3763 | 3727 | 36 | 3734 | 29 |

Table 4 presents observations on several test samples from Bengali and Devanagari datasets to realize the proficiency of SEWM-CNN as well as the cause of the misclassified samples. According to Table 3, although SEWM-CNN outperformed CNN, all the samples misclassified by CNN were not truly classified by SEWM-CNN. Table 4a shows several handwritten numeral images from a test set of Bengali data sets and gives individual numeral-wise classifications by CNN and SEWM-CNN, and remarks on actions. Among six samples, the first three samples were misclassified by CNN, but truly classified by SEWM-CNN, showing the proficiency of SEWM integration with CNN, and the rest of the samples are in the category for those SEWM-CNN also failed to be properly classified. Remarks on three true classified cases by SEWM-CNN are common for both ১ and ৯ numeral cases; CNN's confidence in the wrong classification was below the threshold value (i.e., below 0.6), and SEWM-CNN considered SEWM's decision, which was in the

true numeral category. On the other hand, the reason for SEWM-CNN misclassification is because the decision of SEWM was not considered, as CNN wrongly classified with a high confidence value (Sl. 6) or SEWM agreed to CNN's misclassification (Sl. 4). Observations are also similar for Devanagari samples as shown in Table 4b: SEWM-CNN provides the correct result while SEWM rectified the wrong decision of CNN (Sl. 1–3); SEWM-CNN failed to correctly classify in cases when CNN wrongly classified with a high confidence value (Sl. 6) or SEWM agreed with CNN's wrong classification (Sl. 4 and 5). Both Bengali and Devanagari misclassified samples are very confusing even by human eye judgment. It is notable that the test samples were reserved for checking the proficiency of the system and were not included in any phase of system development. Therefore, the misclassification of several test samples that are much harder to understand even by a human being is acceptable logically.

### 3.2. Performance Comparison

This section compares the performance of the proposed SEWM-CNN with well-known existing works in recognition of Bengali and Devanagari handwritten numerals. Along with test set recognition accuracy, dataset uses and distinguished properties of individual methods are also presented for better understanding in comparison Table 5. Both CNN-based methods and feature-based methods are included in the comparison. Few existing methods are only tested on both scripts. Several feature-based methods used self-prepared datasets, and the number of samples in training and test sets are different from the ISI datasets used in this study. However, the proposed method outperformed any feature-based method for both Bengali and Devanagari. For Bengali, among the feature-based methods, the most recent work with Memory-Based Histogram + GA for feature selection and classification with KNN [35] is shown the best recognition accuracy; the achieved recognition accuracy of 98.40% is inferior to the proposed method. On the other hand, the pioneering work with wavelet filter and classification with a cascade of several MLPs on the ISI datasets [31] is still shown the best recognition accuracy for Devanagari; the achieved recognition accuracy of 99.04% is also inferior to the proposed method.

The proposed SEWM-CNN is with the integration of the SEWM measure with CNN; therefore, its performance comparison with other CNN-based methods is more appropriate. The existing CNN-based methods presented in Table 5 are also tested on the same ISI datasets, which makes the comparison more justified. Among several CNN-based methods for Bengali, the most recent work with rotation-based generated patterns [22] is shown the best recognition accuracy. In [22], two additional training sets are created, rotating original training samples with fixed defined angles clockwise and anti-clockwise; along with the original samples, two different approaches are considered to train CNN. In the case of multiple CNNs, three training sets are used to train three different CNNs (with the same architecture) individually, and the final outcome is generated by combining the decisions of the three CNNs. In another approach, a single CNN is trained, combining the three training sets. The multiple CNN case is shown better performance than the single CNN case, and the achieved recognition accuracy is 98.98% for Bengali. The weakness of the method is reported for the misclassification of similarly shaped numerals (e.g., ১ and ৯) interchangeably. The method investigated in this study tackles the issue of integrating SEWM with CNN and has been shown to achieve an acceptable result. Therefore, the proposed SEWM-CNN outperformed the multiple CNNs cases with a recognition accuracy of 99.20%, even with a single CNN. The work of [22] also investigated Devanagari and achieved recognition accuracies of 99.31% and 98.96% for multiple CNNs and single CNN cases, respectively. The proposed SEWM-CNN is shown a recognition accuracy of 99.23% for Devanagari, and the value is better than a single CNN and competitive with multiple CNNs cases [22]. However, the performance of SEWM-CNN with an ordinary trained CNN (training with available data) is a more remarkable achievement than a CNN training with a three times larger training set [22]. Finally, the achieved recognition accuracy of the proposed SEWM-CNN in comparison with the standard CNN-based methods

(e.g., [23]) and CNN with data augmentation (e.g., [22]) revealed the proposed method as an effective one for recognizing Bengali and Devanagari handwritten numerals.

**Table 4.** Proficiency of SEWM-CNN comparing CNN on selected test samples.

| | Sl. | Handwritten Image | $CL_{CNN}$-$CL_{Sys}$-True Category | Remarks on the Sample. N.B.: $\sigma_0 = 0.6$ |
|---|---|---|---|---|
| **(a) Bengali** | | | | |
| Samples for SEWM rectified CNN's Wrong Decision | 1 | | ৬-৯-৯ | $\sigma_{CNN} = 0.53$ and $CL_{Sys} = CL_{SEWM} = 9$ (i.e., ৯) |
| | 2 | | ৬-৯-৯ | $\sigma_{CNN} = 0.54$ and $CL_{Sys} = CL_{SEWM} = 9$ (i.e., ৯) |
| | 3 | | ৯-১-১ | $\sigma_{CNN} = 0.46$ and $CL_{Sys} = CL_{SEWM} = 1$ (i.e., ১) |
| Samples for SEWM-CNN Misclassified | 4 | | ৯-৯-১ | $\sigma_{CNN} = 0.59$ and $CL_{Sys} = CL_{SEWM} = 9$ (i.e., ৯) |
| | 5 | | ৩-0-৬ | $\sigma_{CNN} = 0.51$ but $CL_{Sys} = CL_{SEWM} = 0$ (i.e., 0) |
| | 6 | | 8-8-৭ | $\sigma_{CNN} = 0.67$, so $CL_{Sys} = CL_{CNN} = 4$ (i.e., ৪) ignoring SEWM |
| **(b) Devanagari** | | | | |
| | Sl. | Handwritten Image | $CL_{CNN}$-$CL_{Sys}$-True Category | Remarks on the Sample. N.B.: $\sigma_0 = 0.6$ |
| Samples for SEWM rectified CNN's Wrong Decision | 1 | | ४-५-५ | $\sigma_{CNN} = 0.52$ and $CL_{Sys} = CL_{SEWM} = 5$ (i.e., ५) |
| | 2 | | ४-५-५ | $\sigma_{CNN} = 0.55$ and $CL_{Sys} = CL_{SEWM} = 5$ (i.e., ५) |
| | 3 | | ५-४-४ | $\sigma_{CNN} = 0.49$ and $CL_{Sys} = CL_{SEWM} = 4$ (i.e., ४) |
| Samples for SEWM-CNN Misclassified | 4 | | ८-८-० | $\sigma_{CNN} = 0.48$ and $CL_{Sys} = CL_{SEWM} = 8$ (i.e., ८) |
| | 5 | | ९–९–६ | $\sigma_{CNN} = 0.54$ and $CL_{Sys} = CL_{SEWM} = 9$ (i.e., ९) |
| | 6 | | ०-०-९ | $\sigma_{CNN} = 0.69$, so $CL_{Sys} = CL_{CNN} = 0$ (i.e., ०) ignoring SEWM |

**Table 5.** Comparison of proposed SEWM-CNN with prominent methods for Bengali and Devanagari HNR in terms of recognition accuracy, dataset used and method's significance.

| Work Reference | Dataset, Ref.; Training and Test Samples | Recognition Accuracy | | Method's Significance in Feature Selection and Classification |
|---|---|---|---|---|
| | | Bengali | Devanagari | |
| Wen et al., 2007 [2] | Postal system; 6000 and 10,000 | 95.05% | - | PCA-based feature selection and SVM for classification. |
| Bhattacharya and Chaudhuri, 2009 [31] | ISI [30]; 19,392 and 4000 | 98.20% | 99.04% | Wavelet filter-based feature selection and cascade of four MLPs for classification. |
| Wen and He, 2012 [44] | Postal system; 30,000 and 15,000 | 96.91% | - | Feature selection using eigenvalues and eigenvectors and classification using kernel and Bayesian discriminant. |
| Das et al., 2012 [3] | CMATERdb 3.1.1 [45]; 4000 and 2000 | 97.70% | - | Feature selection in different stages using GA and classification using SVM. |
| Nasir and Uddin, 2013 [4] | Self-prepared, 300 | 96.80% | - | Bayes' theorem, k-means clustering and Maximum Posteriori for feature selection and SVM for classification. |
| Kumar and Ravulakollu, 2014 [46] | CPAR-2012 [46]; 24,000 and 11,000 | - | 97.87% | Features are based on profile and gradient and classification using NNs (in ensemble and cascade manners) and KNN. |
| Singh et al., 2014 [32] | Samples from ISI [30]; 1400 and 600 | - | 98.53% | Feature selection using information theoretic-based MRMR and classification using NNs and ensemble of NNs. |
| Arya et al., 2015 [33] | ISI [30]; 19,798 and 3763 | - | 98.06% | Feature selection using Gabor filter and classification using KNN and SVM. |
| Singh et al., 2016 [47] | CMATERdb 3.2.1 [45]; 2000 and 1000 | - | 98.92% | Moment based six different features and classification using MLP. |
| Guha et al., 2019 [35] | Self-prepared + Samples from ISI [30]; 10,000 and 500 | 98.40% | 97.60% | Memory-Based Histogram + GA for feature selection and KNN for classification. |
| | | 98.05% | 95.05% | Memory-Based Histogram + GA for feature selection and MLPs for classification. |
| Akhand et al., 2016 [23] | ISI [30]; 18,000 and 4000 | 98.45% | - | Standard CNN |
| Shopon et al., 2016 [48] | ISI [30]; 19,313 and 3986 | 98.29% | - | Auto-encoder (AE) with CNN |
| Akhand et al., 2018 [22] | ISI [30]; 18,000 and 4000 (Bengali)/3763 (Devanagari) | 98.98% | 99.31% | Ensemble of three CNNs; one is trained with available samples and other two used rotation based generated data. |
| | | 98.96% | 98.96% | CNN is trained using available data plus rotation based generated data. |
| Proposed SEWM-CNN | ISI [30]; 18,000 and 4000 (Bengali)/3763 (Devanagari) | 99.20% | 99.23% | Start-end Writing Measure is integrated with CNN's decision. |

## 4. Conclusions

HNR is complex due to inter-class similarity and intra-class difference. Although the convolutional neural network (CNN) is the most successful model for image classification, it suffers from low accuracy in recognizing similarly shaped numerals. This study investigated an innovative HNR method focusing on similarly shaped numerals accompanying a significant property of human writing style. Individual numerals have significant start and end positions, and the start-end writing measure (SEWM) technique and its integration to rectify CNN's decisions are the major contributions of this study. The proposed SEWM-CNN has been tested on Bengali and Devanagari benchmark datasets and is shown to achieve better recognition accuracy than standard CNN.

A number of potential future directions of research can be foreseen out of the present study in the development of a better HNR system. The proposed method is shown to improve performance in altering CNN's decisions when start-end positions are significantly different for similarly shaped numerals (e.g., Bengali ১ and ৯). Taking motivation from this

study, an interesting and challenging task is to improve performance for other numerals with similarities. The idea of a start-end measure in a different form rather than a simple Euclidian distance-based method of this study might be a promising future direction of research. Another challenging task is to develop a different way to rectify the decision of CNN with a different hypothesis rather than a start-end writing measure.

**Author Contributions:** Conceptualization, M.A.H.A.; formal analysis, M.A.H.A., M.R.-U.-Z. and S.H.; funding acquisition, M.A.S.K.; investigation, M.A.H.A. and M.R.-U.-Z.; software, M.R.-U.-Z. and S.H.; supervision, M.A.H.A.; writing—original draft, M.A.H.A., M.R.-U.-Z. and S.H.; writing—review and editing, M.A.H.A. and M.A.S.K. All authors have read and agreed to the published version of the manuscript.

**Funding:** This research received no external funding.

**Data Availability Statement:** Source codes and datasets are available in GitHub repository and the link is https://github.com/AkhandKUET/HNR-integrating-Start-End-Writing-Measure-with-CNN, accessed on 10 January 2023.

**Acknowledgments:** The authors would like to express gratitude to U. Bhattacharya of the Indian Statistical Institute, Kolkata, India, for providing the benchmark datasets used in this study.

**Conflicts of Interest:** The authors declare no conflict of interest.

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
