# Peer review of "Handwritten Numeral Recognition Integrating Start–End Points Measure with Convolutional Neural Network"

_electronics, doi:10.3390/electronics12020472_

Round 1

Reviewer 1 Report

-> This paper discusses and  improved HNR method focusing on similarly shaped numerals, which is interesting for the reader of this paper. This paper also reported with enhanced recognition accuracy demonstrates the superiority of the proposed system over existing methods.

-> Add one dedicated Section on "Related & Background" in this paper with number as 2. It seems that the authors missed recent relevant references on HNR Methods They did not review the state-of-the-art methods published in relative journals or conferences recently (within 2-5 years). Also, add 2-3 tabular comparative analysis for the same in

-> Table 2 on Page#08, should be used as figure as it has lot of sample images.

-> In line 305, add one line diagram/figure to improve the importance of "Significance of the Proposed System".

-> Result section is upto date and appreciated.

->Add more references and cite below article to improve the readability of your paper:

(a) "RVFR: Random Vector Forest Regression Model for Integrated & Enhanced Approach in Forest Fires Predictions", Robin Singh Bhadoria, Manish Kumar Pandey, Pradeep Kundu, Ecological Informatics (Elsevier), Vol. 66, p.101471, 2021

Author Response

Reviewer 1

  1. This paper discusses and improved HNR method focusing on similarly shaped numerals, which is interesting for the reader of this paper. This paper also reported with enhanced recognition accuracy demonstrates the superiority of the proposed system over existing methods.

Response:  We thank the reviewer for appreciating our study and acknowledging our efforts in presenting the work. We have revised the manuscript to comply with the comments and suggestions.

  1. (a) Add one dedicated Section on "Related & Background" in this paper with number as 2. (b) It seems that the authors missed recent relevant references on HNR Methods. They did not review the state-of-the-art methods published in relative journals or conferences recently (within 2-5 years). Also, add 2-3 tabular comparative analysis for the same in

Response: (a) Thanks for the suggestion for adding a new section to describe the related work and background studies. In the original manuscript, we already provided the general background with related work in the introduction section and a review of a specific dataset in the proposed method section, which covers what the reviewer has suggested.

In this revised manuscript, keeping our original structure the same for better readability, we have marked the related work subsection in Section 2.2.1 by revising the contents to address the comments by the reviewer. Note that this part specifically reviews the benchmark datasets and corresponding related work.

 (b) We have searched for the related studies and included several recent studies in the references. The new references are cited accordingly. [Refs. 24-29]

  1. Table 2 on Page#08, should be used as figure as it has lot of sample images.

Response: Thanks for the suggestion. In the revised manuscript contents of the mentioned table are placed in Fig. 4. The indexes of the figures and tables are updated accordingly. [Page 8]

  1. In line 305, add one line diagram/figure to improve the importance of "Significance of the Proposed System".

Response: Thanks for the suggestion. In the manuscript, the proposed model is demonstrated with a graphical representation in Fig. 1, and then individual components are described in individual subsections. The comparative results are demonstrated using several tables and figures too. Therefore, we believe that an additional diagram might be redundant in Section 2.5 in summarizing key significant points of the study. However, to address the reviewer's comments in the revision, the graphical demonstration issue is recalled in the section to improve the paper's readability. [Page 9, Para 2]

  1. Result section is upto date and appreciated.

Response:  Thanks a lot for the insightful evaluation to an appreciation of our study.

  1. Add more references and cite below article to improve the readability of your paper:

(a) "RVFR: Random Vector Forest Regression Model for Integrated & Enhanced Approach in Forest Fires Predictions", Robin Singh Bhadoria, Manish Kumar Pandey, Pradeep Kundu, Ecological Informatics (Elsevier), Vol. 66, p.101471, 2021

Response: We searched the related studies and included several recent studies in the references. The new references are cited accordingly. On the other hand, we failed to link the suggested article with our HNR study as it is not relevant to our work. [Refs. 24-29]

Reviewer 2 Report

1.The handwritten recognition using  CNN is successful applied by many authors especially in english documents for mam-computer communication. What is the aim of presented research (translation to english, spanish or other world languages?)

2. Of course CNN methods together with HNR are well known, the problem is related with new, original aspects of presented paper (Start End Points Measure). the Authors don't describe such algorithm and comparison with methods used by other autors for normalisation and processing the input data. The applied algorithms (both for CNN and HNR) must be described and presebted more detailed together with used methods (criteria of classification an decision).

3. Authors present the tables and charts with high eficiency of recognition (with respect to number of iteration) of  some imput symbols, but more readible are some examples of practical applications. 

Author Response

Reviewer 2

  1. The handwritten recognition using CNN is successful applied by many authors especially in english documents for mam-computer communication. What is the aim of presented research (translation to english, spanish or other world languages?)

Response: We agree with the reviewer that CNN is well-studied for NHR. The weakness of CNN for numerals with similarly shaped patterns has been identified in our study. The aim of the presented research is an enhanced HNR system to improve the classification accuracy of the similarly shaped handwritten numeral beyond the scope of the English Language. The proposed system incorporates the terminal points with CNN’s recognition, which enhances the recognition accuracy in any Language. The secondary aims of such an HNR can be very wide, as the reviewer mentioned (translation to English, Spanish, or other world languages; HNR to voice conversation, and so on), which is beyond the scope of this work. In the manuscript, the aim of this work is mentioned in the abstract and introduction of the revised manuscript. [Page 1, Abstract; Page 2, Para 4]

  1. Of course CNN methods together with HNR are well known, the problem is related with new, original aspects of presented paper (Start End Points Measure). the Authors don't describe such algorithm and comparison with methods used by other autors for normalisation and processing the input data. The applied algorithms (both for CNN and HNR) must be described and presebted more detailed together with used methods (criteria of classification an decision).

Response: Thanks for the appreciation of our study. This paper presents an enhanced HNR system to improve the classification accuracy of the similarly shaped handwritten numerals incorporating the terminals points with CNN’s recognition. Start-End Writing Measure (SEWM) and its integration with CNN is the main contribution of this research. As SEWM is the main attraction of the study, its individual steps, including the classification criteria, are described in Section 2.3. On the other hand, descriptions of general operations on the numeral images are shortened as those are available in the existing studies.  

  1. Authors present the tables and charts with high eficiency of recognition (with respect to number of iteration) of some imput symbols, but more readible are some examples of practical applications.

Response:  Thanks a lot for the insightful evaluation and appreciation of our study.